# Learning Dynamic Protein Representations at Scale with Distograms

**Nicolas Portal[1], Wissam Karroucha[2,3,4], Vincent Mallet[2,3,4,†], Massimiliano Bonomi[1,†]**

[1]Institut Pasteur, Université Paris Cité, CNRS UMR 3528, Computational Structural Biology Unit, Paris, France;
[2]Mines Paris, PSL Research University, CBIO, Paris, France;
[3]Institut Curie, PSL Research University, Paris, France;
[4]INSERM, U1331, Paris, France;

[†] co-corresponding authors.
`vincent.mallet@minesparis.psl.eu`  `massimiliano.bonomi@pasteur.fr`

## Abstract

Protein function and other biological properties often depend on structural dynamics, yet most machine-learning predictors rely on static representations. Physics-based molecular simulations can describe conformational variability but remain computationally prohibitive at scale. Generative models provide a more efficient alternative, though their ability to produce accurate conformational ensembles is still limited. In this work, we bypass expensive simulations by leveraging residue–residue distance probability distributions (distograms) from structure predictors such as AlphaFold2. Our approach provides a scalable way to encode dynamic information into protein representations, aiming to improve function prediction without explicit conformational sampling. All code required to reproduce the experiments presented in this work is publicly available at `https://github.com/nicolas1805961/DistoDyn`.

## 1 Introduction

Proteins perform a wide range of functions within cells. Recently, Machine Learning (ML) approaches have been developed to predict protein function, particularly their interactions with small molecules, RNA, DNA, and other macromolecules. Some methods leverage information from a protein's three-dimensional structure, while others rely on its amino acid sequence to take advantage of the more abundant sequence data. However, since structure encodes function more directly, structure-based methods generally outperform those based solely on sequence Yan et al. (2023).

Structure alone is often not enough to understand protein functions. Biological systems populate a variety of conformational states, and their functions often emerge from the interplay between structural and dynamic properties. For example, conformational transitions Nussinov et al. (2023), allosteric regulation Wodak et al. (2019), and ligand binding site flexibility Alghamedy et al. (2018) play a crucial role to achieve specific functions. Experimental techniques such as nuclear magnetic resonance spectroscopy and cryo-electron microscopy provide valuable insights into biomolecular dynamics but remain limited in their ability to fully characterize complex conformational landscapes at atomistic resolution. Consequently, simulation-based and integrative approaches currently represent the most effective strategies to characterize protein dynamics Hoff et al. (2024).

Simulation methods used to model protein conformational ensembles can be broadly categorized into physics-based approaches and generative models. The main limitation of physics-based methods, such as molecular dynamics (MD) simulations, lies in the prohibitive computational cost required to exhaustively sample complex conformational landscapes. Generative models are also not guaranteed to explore all relevant conformational states, and furthermore their accuracy ultimately depends on the quantity and quality of the data they have been trained on.

ML models aiming at predicting protein function ultimately need to account for the dynamic nature of proteins. While state-of-the-art ML approaches are not built to support multi-conformation in-

puts Hu & Ohue (2025); Cao et al. (2025), recent studies have explored the possibility to encode multiple conformations directly into protein representations. A recent approach aggregates different conformations into pairwise residue correlations, ultimately improving static representations Guo et al. (2025). While promising, the method depends on the availability of extensive MD datasets.

In this work, we take a different approach, completely sidestepping the explicit generation of protein conformations. Specifically, we leverage the probability distributions over residue–residue pairwise distances (distograms) predicted by modern structure prediction methods such as AlphaFold2 Jumper et al. (2021) and Boltz2 Passaro et al. (2025). These distributions have recently been shown to capture prediction uncertainty as well as structural dynamics Brotzakis et al. (2025); Schnapka et al. (2025); Sen et al. (2025); Savaş et al. (2025). Notably, distograms are obtained as byproducts of the structure prediction pipeline and are therefore orders of magnitude cheaper to compute than MD-derived correlation features (Figure 1).

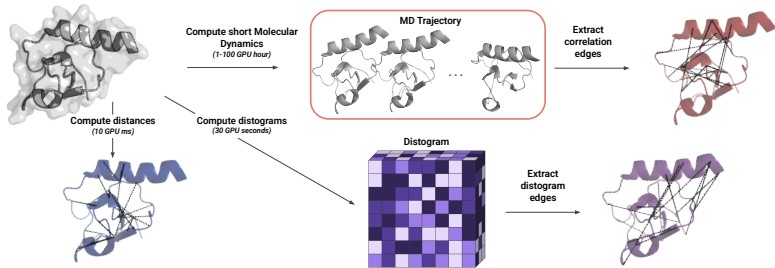

Figure 1: Overview of the protein representations used in this study. A protein can be represented as a static structure, computed on the fly. To capture dynamics, previous work often relies on compute-intensive MD simulations, encoding the results as additional edges. In contrast, we propose bypassing simulations by leveraging predicted distograms to incorporate dynamic information.

## 2    RELATED WORK

**Generating protein conformational ensembles**   Currently, the most popular approach to generate conformational ensembles of proteins and other biomolecules is MD, which draws samples from the Boltzmann distribution given a model of the physico-chemical interactions, or force field. Due to the high computational cost of sampling complex conformational landscapes, several different strategies have been developed. Enhanced sampling techniques Hénin et al. (2022), such as metadynamics Laio & Parrinello (2002), have been developed to accelerate the exploration of conformational landscapes. More recently, large-scale MD datasets have been released Amaro, R. E. et al. (2025); Korlepara et al. (2024); Siebenmorgen et al. (2024); Vander Meersche et al. (2024), opening the door to training generative models directly on MD data.

Due to the expensive cost of MD, generative models have grown attention as a convenient alternative to generate conformational diversity Klein et al. (2023); Jing et al. (2024a); Costa et al. (2024); Lombard et al. (2025); Wolf et al. (2025); Jing et al. (2024b); Cheng et al. (2025), in some cases approximating the Boltzmann distribution Noé et al. (2019); Mardt et al. (2018); Lewis et al. (2025); Tan et al. (2025); Akhound-Sadegh et al. (2025); Lu et al. (2025); Roney et al. (2025). Another class of approaches modify modern structure prediction models to generate conformational diversity, for example by manipulating the multiple sequence alignment del Alamo et al. (2022); Wayment-Steele et al. (2024); Kalakoti & Wallner (2025; 2026) or by steering diffusion for diversity Richman et al. (2025). In some cases, these generative models have been shown to reproduce experimental observables such as NMR order parameters and small-angle X-ray scattering profiles, as well as temperature-dependent ensemble properties Lewis et al. (2025); Janson et al. (2025).

**Protein representation learning**   Geometric deep learning encoders have been used for protein representation learning Isert et al. (2023), using various representations and architectures, such as 3D convolutional networks Jiménez et al. (2017); Weiler et al. (2018), sequence Rao et al. (2021), surfaces Gainza et al. (2020), graphs Aumentado-Armstrong (2018) and equivariant discrete networks Jing et al. (2021). In addition, some methods were developed ad-hoc to handle protein structure,

where protein properties are baked into the network Zhang et al. (2022); Hermosilla et al. (2020); Fan et al. (2022); Wang et al. (2022; 2025).

Multi-modal protein representations can encode different biological and computational priors. A well-studied combination is the use of sequence information along with a graph representation of the structure Hermosilla et al. (2020); Fan et al. (2022); Wu et al. (2023); Zhang et al. (2023). Some approaches include information derived from protein structures in the training of protein language models Bepler & Berger (2019); Heinzinger et al. (2024); Su et al. (2023). More recently, approaches combining different structure representations have demonstrated strong performances Somnath et al. (2021); Mallet et al. (2025); Zhang et al. (2024).

A few methods have emerged to incorporate MD simulations in ML-based representations. Some methods consider different conformations similarly as data augmentation for input protein structures Wu et al. (2022); Min et al. (2024); Libouban et al. (2025). Other approaches adopt a multi-instance learning framework Ilse et al. (2018), encoding each conformation independently and grouping their outputs Zankov et al. (2021); Kleiman et al. (2025). This increases inference time for a limited performance gain Criscuolo et al. (2024). Finally, some approaches directly aggregate the different conformations into a composite graph Chiang et al. (2022); Kalifa et al. (2025); Guo et al. (2025). All these approaches rely on datasets of MD trajectories.

## 3  MOTIVATION AND CONTRIBUTIONS

To encode dynamic information obtained from MD trajectories, Guo et al. (2025) proposed to use residue-residue motion correlation as a pairwise relationship. They enrich the radius graph traditionally used in graph-based protein representation learning, with this additional relationship. Training relational graph neural networks with enriched graphs enhances performance across various tasks.

Here, we follow a similar strategy by extracting information about dynamics from distograms. Distograms are distance distributions between $C_\beta$ atoms (or $C_\alpha$ for glycines). These distributions, first introduced in methods presented at CAPRI13 Senior et al. (2020); Xu & Wang (2019), are now provided by most state-of-the-art structure predictors, such as AlphaFold2 and Boltz2, as a set of equally spaced bins spanning a distance range from 0.2 to 2.2 nm, with the last bin also capturing distances beyond the upper limit. Distograms can be generated at scale for training and deployed at inference time. By leveraging the full probability distribution over inter-node distances, they can capture both spatial proximity and structural uncertainty Brotzakis et al. (2025); Schnapka et al. (2025); Sen et al. (2025); Savaş et al. (2025).

Our key contributions can be summarized as follows:

- We enrich distance-based residue graphs, by extracting edges from distograms, as well as edge features, and encode these graphs with relational graph neural networks.
- We compare our enriched graphs to ones encoding MD trajectories, with enhanced results.
- We successfully apply our protocol to protein and RNA tasks without MD data.

In the following, we present the construction of our graphs and their processing in Section 4. We then compare our approach to the previously proposed approach to encode dynamics using correlations extracted from MD simulations, in cases where MD trajectories are available (Section 5). Finally, we apply our approach to the prediction of protein stability and RNA properties in Section 6. Our method opens the door to function prediction beyond static structures at limited computational cost.

## 4  GRAPH CONSTRUCTION

Protein structures are represented by a graph $\mathcal{G}_{\mathbf{P}} = (\mathcal{V}, \mathcal{E})$. Depending on the task, $\mathcal{V}$ corresponds to either atoms or residues represented by their $C_\alpha$ atom. Node features are defined as one-hot encoding of the amino acid type or atom type. Traditionally, edges connect residues close in three-dimensional space, as determined by k-nearest neighbors or a radius cutoff. In this work, we used radius graph edges, defined as $\mathcal{E}_{dist} = \{(v_i, v_j) \mid d(v_i, v_j) < \tau_{dist}\}$, where $d(v_i, v_j)$ represents the distance between nodes $v_i$ and $v_j$, and $\tau_{dist}$ is a distance threshold.

When an MD trajectory is available, we follow the approach of Guo et al. (2025) and compute the correlation between the motions of residues $v_i$ and $v_j$, denoted as $|C_{ij}|$. We define correlation-based edges $\mathcal{E}_{corr} = \{(v_i, v_j) \mid |C_{ij}| > \tau_{corr}\}$, where $\tau_{corr}$ is a correlation threshold. Following Guo et al. (2025), $\tau_{dist}$ and $\tau_{corr}$ are set to 10Å and 0.3 respectively, (4.5Å and 0.6 for atomic-graphs). Finally, Chroma's developers Ingraham et al. (2023) suggested adding random edges in protein graphs. These edges enable long-range message passing and represent an important negative control for our approach. We sampled edges uniformly, taking as many samples as the number of distogram edges, to obtain $\mathcal{E}_{rand} \sim \mathcal{U}(\mathcal{V} \times \mathcal{V})$, s.t. $|\mathcal{E}_{rand}| = |\mathcal{E}_{dist}|$.

## 4.1 DISTOGRAM-BASED EDGE FEATURES

Distograms $\mathcal{D}$ encode the probability distribution of the distance between each pair of residues $(v_i, v_j)$, effectively representing prediction uncertainty and, possibly, variability due to the underlying dynamics Brotzakis et al. (2025); Schnapka et al. (2025); Sen et al. (2025); Savaş et al. (2025). In practice, these probabilities are discretized into $B$ bins, $(b_1, b_2, \cdots, b_B)$, with $b_1 = 0$ and the convention that $b_{B+1} = \infty$. Distograms are therefore tensors of shape $(N, N, B)$, where $N$ is the number of residues in the graph. The value at position $(i, j, k)$ corresponds to the predicted probability that the distance between nodes $v_i$ and $v_j$ falls into bin $k$, i.e. $\mathcal{D}(i, j, k) = \mathbb{P}[b_k \leq d(v_i, v_j) < b_{k+1}]$.

Based on these probabilities, we can define a distogram-based edge set composed of pairs predicted to be close with sufficient probability,

$$\mathcal{E}_{\text{disto}} = \{(v_i, v_j) \mid \mathbb{P}[d(v_i, v_j) \leq \delta] > \tau_{\text{disto}}\}, \tag{1}$$

where $\tau_{\text{disto}}$ is a probability threshold and $\delta$ is a distance cutoff for the distogram-based neighborhood. The value of $\delta$ is determined from the distribution of distances between neighboring residues as measured in the Protein Data Bank (PDB), and depends on the specific pairs $(u, v)$ of amino-acid types as $\delta_{u,v} = \mu_{u,v} + 1.645\sigma_{u,v}$ where $\mu_{u,v}$ and $\sigma_{u,v}$ are tabulated for each amino-acid pair Kamisetty et al. (2013). To be used in the discrete setting of distograms, $\delta_{u,v}$ is translated into a cutoff bin defined as the bin closest to the cutoff distance, $b_\delta^{uv} = \text{argmin}_{b \in \{b_1, \ldots, b_B\}} d(b, \delta_{uv})$.

The probability on the left hand size of Eq. 1 is computed as $\mathbb{P}[d(v_i, v_j) \leq \delta_{v_i, v_j}] = \sum_{k \leq b_\delta^{v_i, v_j}} \mathcal{D}(i, j, k)$. In our experiments, distograms were extracted from the confidence head of Boltz2 Passaro et al. (2025) after softmax normalization. Our results were obtained with $\tau_{\text{disto}} = 10^{-4}$, which corresponds to roughly the number of edges computed from MD correlations. In addition to enriching graphs with additional edges, we can use the probability distributions $\mathcal{D}(i, j) \in \Delta^B$ as edge features. This allows the model to capture both the expected distances and the dynamic variability between nodes.

## 4.2 GRAPH ANALYSIS

We analyze the different edge types introduced above using the MISATO dataset Siebenmorgen et al. (2024), a collection of thousands of short MD simulations of protein-ligand complexes. A detailed description of this dataset is provided in Appendix A.2. We first compute the intersection of the different edge types, counted as a fraction of the total number of possible edges (Figure 2A). There is a fair agreement between edge types, and notably distance edges (3.1% of possible pairs) are covered by the distogram and correlation edges (only 0.1% are specific to distance). We notice a moderate overlap between correlation edges and distogram edges.

To illustrate their difference, we use the `6nvd` protein, an enzyme involved in biotin synthesis in the bacteria responsible for tuberculosis. First, we show the edge lengths for each type in Figure 2C, and observe increased means for $\mathcal{E}_{corr}$ (9.63Å) and $\mathcal{E}_{disto}$ (9.57Å) compared to $\mathcal{E}_{dist}$ (7.16Å). This trend is even stronger on the whole dataset, with means values of 17.1Å, 19.2Å and 7.2Å, respectively. Hence, correlation-based and distogram-based edges both enable long-distance message passing.

Moreover, we notice that these edges have distinct patterns (Figure 2D). Notably, edges specific to correlation were mapped to the structure (Figure 2B, in red). These edges correspond to a helix and a beta-sheet motif that move in a correlated way. Therefore, the corresponding edges do not correspond to proximity in alternative conformations. On the other hand, distogram edges are found around distance edges, encoding fuzziness. We also identified a set of distogram-based edges (Figure 2B, in purple), which corresponds to a flexible loop with a highly dynamic secondary structure.

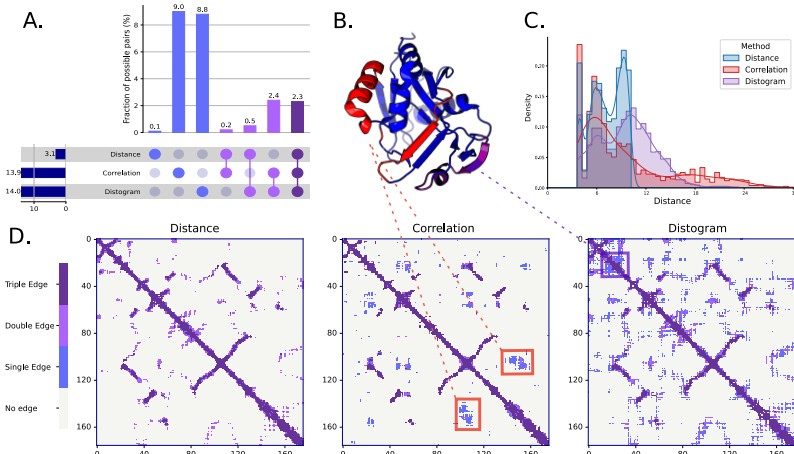

Figure 2: **A.** Upset plot showing the overlap between our different edge types ($\mathcal{E}_{dist}$, $\mathcal{E}_{corr}$, and $\mathcal{E}_{disto}$) across the ligand binding site task test set. Rows represent individual edge types, while columns indicate specific combinations. **B.** 3D structure of the `6nvd` protein (from our test set). Red residues correspond to the red box in the correlation adjacency matrix, and purple residues correspond to the box in the distogram adjacency matrix. **C.** Distance distributions for the different edge types introduced for the `6nvd` protein. **D.** Adjacency matrices for each edge type in the `6nvd` protein. Colors indicate whether a residue pair appears as a single, double, or triple edge—that is, whether it is present in one, two, or all three edge sets ($\mathcal{E}_{dist}$, $\mathcal{E}_{corr}$, $\mathcal{E}_{disto}$).

## 4.3 RELATIONAL GRAPH NEURAL NETWORKS

To effectively leverage the diverse types of edge features described above (distance, correlation, and distogram-based), we employ *relational graph neural networks* (R-GNNs) that are designed to handle graphs with multiple relation types. In this framework, each edge type corresponds to a distinct relation, allowing the model to learn edge-specific message passing rules and therefore capture the different structural and dynamic properties encoded by each edge feature.

We experiment with three R-GNN variants: Relational Graph Convolutional Networks (R-GCN) Schlichtkrull et al. (2018), Relational Graph Attention Networks (R-GAT) Busbridge et al. (2019), and a Relational version of the Equivariant Graph Neural Network (R-EGNN) Satorras et al. (2021). A detailed description of these models is provided in Appendix A.1. By using these relational GNN architectures, the model can capture the complementary information encoded in different edge types, while simultaneously leveraging node features to predict properties at the residue or atom level.

## 5 VALIDATION ON MD DATASETS

We now investigate the impact of encoding distogram information with relational graph networks. We start with scenarios where explicit dynamic information is available in the form of short MD trajectories, to compare to approaches encoding dynamics using $\mathcal{E}_{corr}$.

**Experimental setup** Experiments are conducted on two separate tasks: ligand binding site and ligand binding affinity prediction, as proposed in Guo et al. (2025). Following their work, binding sites are composed of protein residues closer than 10Å to any non-hydrogen atom of the ligand. Entire protein structures are used, resulting in graphs with an average number of nodes equal to 443.

For the binding affinity prediction task, only atoms belonging to the binding pocket are considered, resulting in much smaller graphs (47 residues on average). Moreover, since this task requires representing a ligand, we need to adapt the distogram edge set $\mathcal{E}_{disto}$ introduced in the previous section. Namely, we complement $\mathcal{E}_{disto}$ with atomic radius graph edges on the ligand side, and we connect protein and ligand nodes if one atom of the protein is closer than 8Å to an atom of the ligand.

The three models described above are trained and evaluated on each task using distance, correlation, and distogram-based edges, either individually or in combination as distinct relation types. When using distogram-based edges, the R-GAT and R-EGNN models are also trained with all distograms used as edge features. For both tasks, model architectures and hyperparameters are kept identical across different relation combinations, except for the dropout rate, which is independently tuned for each model to achieve optimal performance. Details about the learning and architecture hyperparameters are provided in Appendix A.3. The results of the R-EGNN model are presented in Table 1 while performance for the R-GCN and R-GAT models is available in the Appendix A.5.

| Binding Site Prediction (Mean number of nodes = 443) | | | | |
|---|---|---|---|---|
| Graph Type | Accuracy | Precision | Recall | F1 score |
| Distance | 0.832 | 0.282 | 0.444 | 0.345 |
| Distance + Correlation | **0.882** | **0.393** | 0.350 | 0.370 |
| Distance + Random Edges | 0.830 | 0.316 | **0.607** | 0.416 |
| Distance + Distogram | 0.861 | 0.376 | 0.606 | **0.464** |
| Distance + Distogram + Features | 0.859 | 0.371 | 0.602 | 0.459 |
| Binding Affinity Prediction (Mean number of nodes = 47) | | | | |
| Graph Type | MAE | RMSE | Pearson R | Spearman R |
| Distance | 1.296 | 1.623 | 0.666 | 0.642 |
| Distance + Correlation | 1.357 | 1.713 | 0.611 | 0.576 |
| Distance + Random Edges | 1.211 | 1.502 | 0.721 | 0.698 |
| Distance + Distogram | 1.275 | 1.560 | 0.699 | 0.674 |
| Distance + Distogram + Features | **1.208** | **1.479** | **0.736** | **0.725** |

Table 1: Results on the binding site prediction task (Top, average of 443 nodes) and binding affinity prediction task (Bottom, 47 nodes on average) for the R-EGNN model. We compare various dynamic-encoding approaches using $\mathcal{E}_{dist}$ alone or combined with $\mathcal{E}_{corr}$, $\mathcal{E}_{random}$, or $\mathcal{E}_{disto}$. Distogram-based edge features are also incorporated where compatible. Best-performing models are shown in bold, and second-best are underlined.

**Overall performance of our approach**   Incorporating distogram-based edges and features consistently and significantly improves performance across tasks and architectures. Across tasks and models, using distogram always ranks first (16/24 settings) or second. Importantly, when ranking second, it comes as a close second, trailing only 2 accuracy points for R-GCN, but when ranking first it can induce significant boosts (such as 9.3 F1 points or 4 Spearman points on affinity for R-GAT).

**Impact of distogram edges**   On the binding site task, which involves a larger number of nodes, introducing additional edges beside those based on distance brings significant benefits. Moreover, selecting the right edges plays an important role and we observe the following performance (informal) ordering $\mathcal{E}_{random} < \mathcal{E}_{corr} < \mathcal{E}_{disto}$. However, on the binding affinity task the impact of adding distogram edges is more nuanced. While our approach always represents an improvement over using $\mathcal{E}_{dist}$ only, this improvement is comparable to incorporating random edges. Both approaches outperform graphs that use $\mathcal{E}_{corr}$. We attribute this result to the limited size of the binding pockets (47 residues on average), which are already well-connected.

**Impact of distogram edge features**   Incorporating distograms as edge features often results in clear performance improvements (Affinity, R-EGNN, +5 Spearman points), but in some case it can be negligible, or even detrimental. R-GAT benefits from including distogram edge features on the binding site task and suffers on the binding affinity task, while the opposite is true for R-EGNN. However, being able to use one or the other results in a clear improvement in all cases (their performance is not equivalent). We advise users to test both approaches for their specific problem.

**Comparison to MD-correlation based approaches**   Finally, we compare our method to the correlation-based approach proposed by Guo et al. (2025), which relies on compute-intensive MD simulations. Our distogram-based approach clearly outperforms theirs, even without edge features. On the binding site task, distogram edges outperform correlation edges 9 out of 12 times, in some

cases with a substantial gap (*R-EGNN Recall*, 60 vs 35). On the binding affinity task, distogram edges were superior *across all models and metrics considered*.

It should be noted that we do not fully reproduce the results of Guo et al. (2025) that reported consistent improvement by adding correlation edges. In our experiments, the performance of distance-based models was much higher than the reported one, and adding correlation edges did not always improve performance. This was observed on the recall of the binding site task, and more generally on the binding affinity task. Moreover, while adding correlation edges was useful to predict binding sites, we found it to be less efficient than adding random edges to the graph. Overall, our results indicate that distograms capture complementary structural information that is not encoded by neither static distance thresholds nor correlation-based neighborhoods. This added information brings benefits to both residue-level classification and atom-level regression tasks.

# 6 APPLICATIONS ON GENERIC DATASETS

In this section, we evaluate the performance of our approach in predicting the effects of protein mutations and various RNA properties. In both tasks, dynamics play a key role. However, MD simulations datasets are not available, which makes these tasks particularly challenging and highlights the need for accurate prediction methods that do not rely on such data.

## 6.1 PREDICTION OF MUTATION EFFECTS

Protein stability is measured as the change in free energy $\Delta G$ between its folded and unfolded states. Upon mutations in the amino acid sequence, stability can be significantly affected. This variation is quantified by $\Delta\Delta G$, which can be used to compare the effect of different mutations, notably for tasks like protein design. As for other biological properties, dynamics in both folded and unfolded states plays an important role in determining protein stability Frellsen et al. (2025).

We applied our approach to the classic ThermoMPNN architecture Dieckhaus et al. (2024), which is designed to predict $\Delta\Delta G$ from the native structure and mutated sequence. This model is trained on the mega-scale dataset Tsuboyama et al. (2023), a large dataset of experimentally measured $\Delta G$ for relatively small proteins. ThermoMPNN relies on ProteinMPNN Dauparas et al. (2022), a popular graph-based structure encoder, to extract structure embeddings. In our experiments, we sought to enrich the ProteinMPNN graphs with distograms. Data and splits were held constant. Details about the training datasets, architectures, and procedures are reported in Appendix A.3.

Given the small size of the proteins studied here (56 nodes on average) and the high number of neighbors used by ProteinMPNN (48 nearest neighbors), little room is left for adding edges. Therefore, we only investigate adding edge features to the baseline model. In addition, we present an ablation experiment where only 16 neighbors are considered, so that additional edges can be introduced in the graph. We present the results of our experiments in Table 2.

| Connectivity | Distogram Features | $R^2$ | RMSE | Spearman R | Pearson R |
|---|---|---|---|---|---|
| Baseline (Topk = 48) | | 0.518 | 0.727 | 0.726 | 0.761 |
| Baseline | ✓ | **0.577** | **0.681** | **0.749** | **0.788** |
| Ablation (Topk = 16) | ✓ | 0.550 | 0.702 | 0.727 | 0.770 |
| Ablation + 8 Random Edges | ✓ | 0.556 | 0.698 | 0.733 | 0.773 |
| Ablation + 8 Distogram Edges | ✓ | **0.557** | **0.697** | **0.740** | **0.778** |

Table 2: Protein stability results. Best models in bold, second-best underlined.

Our results show that incorporating distogram features clearly improves performance across all metrics. The $R^2$ increases from 0.518 to 0.577, RMSE decreases from 0.7266 to 0.6808, and both Pearson and Spearman correlations improve. This indicates that distograms capture structural variability relevant for stability prediction, providing informative edge features that complement ProteinMPNN. Moreover, when using a reduced set of edges in the ablation experiment, introducing additional edges improve the results. This particularly holds for the Spearman correlation that increases from 0.727 to 0.733 with random edges, and to 0.740 with edges derived from distograms.

## 6.2 Prediction of RNA properties

Finally, we apply our approach to the prediction of various RNA properties. RNA molecules typically display greater flexibility than proteins due to their less hydrophobic nature. We first adapt the graph construction introduced for proteins, such that nodes represent nucleotides instead of amino acids. We apply our approach to the RNA-CM and RNA-Site tasks, introduced in the RNAglib benchmark Wyss et al. (2025). RNA nucleotides can be chemically modified, subtly altering their shape but crucially affecting their functions. The RNA-CM task aims to predict these chemical modifications from an RNA structure. The RNA-Site task is similar to the aforementioned ligand binding site prediction in proteins. These tasks have 57 and 64 nodes on average, respectively.

Considering the limited size of the RNA graphs, we only investigate the impact of adding distogram edge features to RNA graphs. To isolate the impact of dynamics captured by the distogram from static distance information, we systematically report the metrics obtained using distance features (a Gaussian radial basis function encoding of the pairwise residue distances using the same distance bins as those used in the distogram). We report the results across three different graph constructions: distance-based graph (a graph in which each residue node is connected to the nodes of its neighbors in Euclidean space), 2D+ graph (encoding backbone and canonical base pairs), and 2.5D graph (encoding backbone, canonical, and noncanonical base pairs, as defined in Leontis & Westhof (1998)). The results are presented in Table 3, and additional details are provided in Appendix A.4.

| Connectivity | RNA-CM | | | RNA-Site | | |
|---|---|---|---|---|---|---|
| | No Features | Distance Features | Distogram Features | No Features | Distance Features | Distogram Features |
| Distance | 52.8 | 52.2 | **54.8** | 59.2 | **61.9** | 59.3 |
| 2D+ | 64.5 | **65.5** | **65.5** | 61.1 | 59.8 | **62.1** |
| 2.5D | 66.7 | 64.6 | **67.3** | 60.7 | 60.4 | **60.8** |

Table 3: Results on RNA-CM and RNA-Site tasks (metric reported: balanced accuracy).

Adding distogram edge features results in consistent improvements across different graph constructions and tasks. In particular, the superior performance of distogram edge features over distance edge features underscores the ability of distograms to capture valuable information regarding RNA conformational variability. These results are particularly interesting given that state-of-the-art structure predictors are generally less accurate for RNA molecules than for proteins Kretsch et al. (2026), partly owing to the difficulty of tackling RNA flexibility.

## 7 Discussion and conclusions

In this paper, we propose an efficient approach to encode proteins beyond a static structure by using distograms derived from Boltz2, a state-of-the-art protein structure predictor. We propose to enrich graph-based models representing protein structures with edges between residues predicted to be in close proximity according to their corresponding distogram. When the model is compatible, we also include the full distogram distribution as edge features.

For ligand binding site and binding affinity predictions, our approach outperforms methods that rely on computationally expensive molecular dynamics MD simulations. For the prediction of protein stability and RNA properties, where MD data were not available, our method also consistently improved performance. Most importantly, across all tasks, the overall best-performing method among different architectures is the one incorporating distograms.

These results highlight the potential of incorporating distogram-derived information to enhance protein as well as RNA representation learning. Nonetheless, our approach presents a few limitations. First, it relies on high-quality distograms, whose accuracy ultimately depends on the strength of coevolutionary signals in the multiple sequence alignment. Second, computing distograms for large proteins or macromolecular complexes can be computationally demanding; consequently, our analysis was restricted to relatively small systems. Finally, because distograms represent marginalized pairwise distance distributions, they provide an entangled, collective description of conformational variability and may not explicitly capture higher-order, multi-residue dynamic correlations.

An interesting future direction is the application of our method to systems lacking experimental structural information. This setting is particularly challenging, as the performance of structure prediction methods strongly depends on coevolutionary signals and the availability of related structures in the PDB. Consequently, it remains uncertain whether reliable and informative features can be extracted for systems with low-accuracy structural predictions. Nevertheless, our approach represents a promising step toward bridging the gap between the abundance of sequence data and the capabilities of dynamic structural modeling.

MEANINGFULNESS STATEMENT

Protein and RNA are inherently dynamic, yet most computational predictors of their function rely on static structures due to the high cost of molecular simulations. We present a scalable alternative that leverages distograms produced by modern structure predictors to encode dynamic information without explicit simulations. By integrating this information into graph-based models, our approach improves predictions across multiple biological tasks while remaining computationally efficient. This work lowers the barrier to incorporating dynamics into biomolecular machine learning and broadens access to function prediction methods beyond settings where molecular dynamics data are available.

ACKNOWLEDGMENTS

This project has received funding from the European Research Council (ERC) under the European Union's Horizon 2020 research and innovation program (Grant agreement No. 101086685 – bAIes). V.M. is supported by a Junior Springboard Prairie program, funded by the ANR project ANR-23-IACL-0008. W.K. is supported by Fondation pour la Recherche Médicale (FRM) with the following grant number: ECO202406019160. This work was performed using the Maestro cluster at Institut Pasteur and HPC resources from GENCI–IDRIS (Grant AD010315435R1).

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

# A APPENDIX

## A.1 DETAILED MODEL DEFINITION

**Relational Graph Convolutional Network (R-GCN)** R-GCN extends the standard GCN by introducing relation-specific weight matrices. Let $h_i^{(l)}$ be the representation of node $i$ at layer $l$. This representation is modulated by relation-specific weight matrices $W_r^{(l)}$, with the convention that $r = 0$ corresponds to self-loops. The message sent by a node $i$ across relation $r$ is defined as

$$m_{r,i}^l = W_r^{(l)} h_i^{(l)}. \tag{2}$$

Message passing updates the node representation as:

$$h_i^{(l+1)} = \sigma\left(m_{0,i}^{(l)} + \sum_{r \in \mathcal{R}} \sum_{j \in \mathcal{N}_i^r} \frac{1}{c_{i,r}} m_{r,j}^{(l)}\right), \tag{3}$$

where $\mathcal{N}_i^r$ is the set of neighbors of node $i$ under relation $r$, $c_{i,r}$ is a normalization constant, and $\sigma$ is a non-linear activation function.

**Relational Graph Attention Network (R-GAT)**   R-GAT introduces relation-specific attention coefficients to weight messages from different neighbors:

$$h_i^{(l+1)} = \sigma\left(\sum_{r \in \mathcal{R}} \sum_{j \in \mathcal{N}_i^r} \alpha_{ij}^r m_{r,j}^{(l)}\right), \tag{4}$$

where the attention coefficients $\alpha_{ij}^r$ are computed as

$$\alpha_{ij}^r = \frac{\exp\left(\text{LeakyReLU}\left(a_r^\top [m_{r,i}^{(l)} \,\|\, m_{r,j}^{(l)}]\right)\right)}{\sum_{k \in \mathcal{N}_i^r} \exp\left(\text{LeakyReLU}\left(a_r^\top [m_{r,i}^{(l)} \,\|\, m_{r,k}^{(l)}]\right)\right)}, \tag{5}$$

with $a_r$ as a learnable attention vector for relation $r$ and $\|$ denoting vector concatenation.

**Relational Equivariant Graph Neural Network (R-EGNN)**   To extend EGNN to handle multiple edge types, we create a relational variant (R-EGNN) by processing each relation type separately. Specifically, for each edge type $r$ (e.g., distance-based, correlation-based, or distogram-based), a separate EGNN processes the corresponding subgraph, producing relation-specific node and coordinate updates. The outputs from all relations are then combined via summation to obtain the final node embeddings and coordinate updates.

Formally, the node update for relation $r$ follows the standard EGNN message passing:

$$m_{ij}^r = \phi_e(h_i^{(l)}, h_j^{(l)}, \|x_i^{(l)} - x_j^{(l)}\|^2, e_{ij}^r), \tag{6}$$

$$\Delta x_i^r = \frac{1}{|\mathcal{N}_i^r|} \sum_{j \in \mathcal{N}_i^r} (x_i^{(l)} - x_j^{(l)}) \phi_x(m_{ij}^r), \tag{7}$$

$$h_i^{(l+1),r} = \phi_h\left(h_i^{(l)}, \sum_{j \in \mathcal{N}_i^r} m_{ij}^r\right), \tag{8}$$

where $h_i^{(l)}$ and $x_i^{(l)}$ are the feature vector and coordinates of node $i$ at layer $l$, $e_{ij}^r$ is the edge feature vector for relation $r$, and $\phi_e$, $\phi_x$, $\phi_h$ are learnable MLPs.

The final node representation and coordinates are obtained by summing the contributions across all relations:

$$h_i^{(l+1)} = \sum_{r \in \mathcal{R}} h_i^{(l+1),r}, \tag{9}$$

$$x_i^{(l+1)} = x_i^{(l)} + \sum_{r \in \mathcal{R}} \Delta x_i^r. \tag{10}$$

This design allows the model to leverage relation-specific information while maintaining the equivariance properties of EGNN: the predictions are invariant to translations and equivariant to rotations of the input coordinates. By incorporating multiple edge types (distance, correlation, and distogram), R-EGNN captures complementary structural and dynamic information while directly processing 3D node coordinates.

A.2   DATASETS

The MISATO dataset Siebenmorgen et al. (2024) is used for the binding site and binding affinity prediction task. This dataset contains 19443 protein-ligand complexes extracted from PDBbind Su

et al. (2018); Liu et al. (2017); Wang et al. (2005). In the binding site prediction task, the data is partitioned following the split provided with the MISATO dataset. In the binding affinity prediction task, models are trained on the PDBbind 2020 refined set and tested on the core set which contain respectively 5318 and 285 protein–ligand complexes retained after extensive filtering to ensure the quality of both binding affinity data and crystal structures.

Experiments for the protein stability prediction task are conducted on the mega-scale dataset Tsuboyama et al. (2023). This dataset contains 776000 high-quality folding stability measurements spanning all single amino acid variants and selected double mutants across 331 natural and 148 de novo–designed protein domains, each 40–72 amino acids long. After removing duplicate and unreliable stability measurements, we end up with 577313 samples. The data are split into training, validation, and test sets following the partitioning scheme used by Dieckhaus et al. (2024). After removing double mutants, the final dataset contains 298 wild types, accounting for 443,906 protein sequences.

## A.3 IMPLEMENTATION DETAILS

The code necessary to run our experiments and reproduce our results can be found at `https://anonymous.4open.science/r/DistoDyn-55AB/README.md`.

Our distograms were generated using Boltz with default parameters. The MSAs were generated on the fly. For RNA, Boltz does not recommend using MSAs. Alignments were generated with the following command line:

```
boltz predict "${INPUT_PATH}" --use_msa_server --model boltz2
    --out_dir "${OUTPUT_DIR}"
```

For the binding site and binding affinity prediction, we used the models as defined by Guo et al. (2025), but we grid-searched dropout rates over the following values : $\{0.0, 0.1, 0.2, 0.3, 0.4\}$ for all models.

For the finetuning of the protein stability tasks, all layers are kept frozen except the initial edge embedding layers, light attention layers and heads. We chose to fine-tune ThermoMPNN instead of training it from scratch. To do so, we kept the ProteinMPNN encoder layers frozen in early epochs and gradually unfroze them. Out of fairness, we also tried to fine-tune their model (allowing for more computations), which ultimately did not affect performance. Different learning rate scheduling are used for these layers. We tried training our models with 6 different learning rate strategies detailed in Table 4. We report the best test performance. In addition, the first encoder layer of ProteinMPNN is unfrozen after epoch 10. Then between epoch 10 and 20 the second encoder layer of ProteinMPNN is also unfrozen. Finally, the last encoder layer is unfrozen from epoch 20 until the end of training.

| Learning rate setup | | A | B | C | D | E | F |
|---|---|---|---|---|---|---|---|
| Encoder | Initial | $10^{-4}$ | $10^{-4}$ | $10^{-5}$ | $10^{-4}$ | $10^{-3}$ | $10^{-3}$ |
| | Final | $10^{-5}$ | $10^{-5}$ | $10^{-6}$ | $10^{-5}$ | $10^{-4}$ | $10^{-5}$ |
| Edge | Initial | $10^{-3}$ | $10^{-3}$ | $10^{-4}$ | $10^{-4}$ | $10^{-3}$ | $10^{-3}$ |
| | Final | $10^{-5}$ | $10^{-4}$ | $10^{-5}$ | $10^{-5}$ | $10^{-4}$ | $10^{-4}$ |
| Head | Initial | $10^{-3}$ | $10^{-3}$ | $10^{-4}$ | $10^{-4}$ | $10^{-3}$ | $10^{-3}$ |
| | Final | $10^{-5}$ | $10^{-4}$ | $10^{-5}$ | $10^{-5}$ | $10^{-4}$ | $10^{-4}$ |
| Light Attention | Initial | $10^{-3}$ | $10^{-3}$ | $10^{-4}$ | $10^{-4}$ | $10^{-3}$ | $10^{-3}$ |
| | Final | $10^{-5}$ | $10^{-4}$ | $10^{-5}$ | $10^{-5}$ | $10^{-4}$ | $10^{-4}$ |

Table 4: Setups used for the protein stability prediction task

## A.4 RNA SETTINGS

### A.4.1 DETAILS ABOUT GRAPH CONSTRUCTION

We follow the graph nomenclature by Wyss et al. (2025). Three families of graph representations are being benchmarked: distance-based graphs, 2D+ graphs and 2.5D graphs. In all graph representations used, a graph denotes a connected component of an RNA, and a node denotes a residue.

In distance-based graphs, each residue is connected to all residues located within a neighborhood of radius 8.0Å (self-loops are removed). In order to compute residue coordinates, following Boltz2 Passaro et al. (2025) and AlphaFold 3 Abramson et al. (2024), we choose the C2 atom of the base as representative atom of residues with pyrimidine bases and the C4 atom of the base as representative atom of residues with purine bases. This parameterization has the advantage of aligning with the atoms used in the distogram computation by Boltz2 and capturing information regarding base pairing and base stacking, two driving forces of RNA structure and interactions with proteins and small molecules. The 8.0Å cutoff was chosen after a careful examination of graph connectivities for various possible values. In our datasets, when using an 8.0Å cutoff for graph construction, each node has on average 7.7 neighbors, which is reasonable given the relatively small size of our graphs (57 and 64 nodes on average for RNA-CM and RNA-Site Wyss et al. (2025), respectively). In this setting, we note that there is only one edge type.

In 2D+ graphs, three distinct edge types are added. A first edge type is created for 5' to 3' backbone connections (phosphodiester bonds), a second edge type is added for 3' to 5' backbone connections, and a third edge type (bidirectional) for canonical base pairs. These graphs are therefore directed.

In 2.5D graphs, 20 edge types are created: 5' to 3' backbone connections, 3' to 5' backbone connections, and one for each of the base pair geometries as defined by the Leontis-Westhof nomenclature of base pairs Leontis & Westhof (1998). These edges were shown to improve the performance of machine learning approaches for drug design applications Oliver et al. (2020), and on a recent benchmark for RNA 3D structure-function modeling Wyss et al. (2025).

For each of these settings, we run experiments without any edge features, with distogram-based edge features, and with distance-based edge features. In the setting without edge features, the RGAT acts upon edge types and node features only. In the setting with edge features, the RGAT processes edge features, in addition to node features and edge types. Edge features are processed within the attention computation mechanism of the RGAT layer.

For each edge of the graph connecting any two residues, its distogram-based edge features are the probabilities of the pairwise residue distances being in each of the 64 distance bins of the distogram. Its distance-based edge features are the encoding of the pairwise residue distances through Gaussian radial basis functions using the exact same distance bins as Boltz's distograms.

### A.4.2 ARCHITECTURE AND TRAINING HYPERPARAMETERS

The experiments were carried out using relational graph attention networks (RGAT). This architecture was chosen for its ability to natively handle simultaneously distinct edge types and continuous edge features. For a fair comparison across settings, we also used the RGAT architecture for the experiments carried out without distance- or distogram-based edge features. The implementation was performed using RNAglib Tasks Wyss et al. (2025).

The hyperparameters chosen are reported below:

The loss function is weighted by the relative occurrences of the classes in our datasets.

All results reported in Table 3 represent the mean performance computed across three independent trials with distinct random seeds.

Table 5: Hyperparameter settings used for the experimental evaluation on RNA structures

| Hyperparameter | Value |
|---|---|
| Number of Layers | 3 |
| Hidden Channels | 128 |
| Training Epochs | 200 |
| Batch Size | 8 |
| Dropout Rate | 0.5 |
| Loss Weights | Ratio-based |
| Learning Rate | $\{10^{-3}, 10^{-4}\}$ (Tuned via validation) |

## A.5 FULL MD RESULTS

| Binding Site Prediction (Mean number of nodes = 443) | | | | | |
|---|---|---|---|---|---|
| Model | Graph Type | Accuracy ($\uparrow$) | Precision ($\uparrow$) | Recall ($\uparrow$) | F1 score ($\uparrow$) |
| R-GCN | Distance | 0.761 | 0.187 | 0.420 | 0.259 |
| | Distance + Correlation | **0.809** | 0.217 | 0.357 | 0.270 |
| | Distance + Random Edges | 0.773 | 0.193 | 0.405 | 0.261 |
| | Distance + Distogram | 0.782 | **0.236** | **0.535** | **0.328** |
| R-GAT | Distance | 0.753 | 0.196 | 0.478 | 0.278 |
| | Distance + Correlation | 0.791 | 0.229 | 0.469 | 0.308 |
| | Distance + Random Edges | 0.760 | 0.217 | 0.546 | 0.311 |
| | Distance + Distogram | 0.814 | 0.289 | 0.598 | 0.390 |
| | Distance + Distogram + Features | **0.816** | **0.298** | **0.627** | **0.404** |
| R-EGNN | Distance | 0.832 | 0.282 | 0.444 | 0.345 |
| | Distance + Correlation | **0.882** | **0.393** | 0.350 | 0.370 |
| | Distance + Random Edges | 0.830 | 0.316 | **0.607** | 0.416 |
| | Distance + Distogram | 0.861 | 0.376 | 0.606 | **0.464** |
| | Distance + Distogram + Features | 0.859 | 0.371 | 0.602 | 0.459 |
| Binding Affinity Prediction (Mean number of nodes = 47) | | | | | |
| Model | Graph Type | MAE ($\downarrow$) | RMSE ($\downarrow$) | Pearson R ($\uparrow$) | Spearman R ($\uparrow$) |
| R-GCN | Distance | 1.244 | 1.562 | 0.689 | 0.656 |
| | Distance + Correlation | 1.299 | 1.601 | 0.673 | 0.637 |
| | Distance + Random Edges | **1.171** | **1.453** | **0.744** | **0.718** |
| | Distance + Distogram | 1.195 | 1.487 | 0.731 | 0.694 |
| R-GAT | Distance | 1.226 | 1.562 | 0.691 | 0.658 |
| | Distance + Correlation | 1.280 | 1.568 | 0.686 | 0.656 |
| | Distance + Random Edges | 1.236 | 1.542 | 0.701 | 0.685 |
| | Distance + Distogram | **1.176** | **1.487** | **0.740** | **0.713** |
| | Distance + Distogram + Features | 1.179 | 1.492 | 0.722 | 0.693 |
| R-EGNN | Distance | 1.296 | 1.623 | 0.666 | 0.642 |
| | Distance + Correlation | 1.357 | 1.713 | 0.611 | 0.576 |
| | Distance + Random Edges | 1.211 | 1.502 | 0.721 | 0.698 |
| | Distance + Distogram | 1.275 | 1.560 | 0.699 | 0.674 |
| | Distance + Distogram + Features | **1.208** | **1.479** | **0.736** | **0.725** |

Table 6: Results on the binding site prediction task (Top, average of 443 nodes) and binding affinity prediction task (Bottom, 47 nodes on average). We compare various dynamic-encoding approaches using $\mathcal{E}_{dist}$ alone or combined with $\mathcal{E}_{corr}$, $\mathcal{E}_{random}$, or $\mathcal{E}_{disto}$. Distogram-based edge features are also incorporated where compatible. Best-performing models are shown in bold, and second-best are underlined.

| Binding Site Prediction (Mean number of nodes = 443) | | | | | |
|---|---|---|---|---|---|
| Model | Graph Type | Accuracy ($\uparrow$) | Precision ($\uparrow$) | Recall ($\uparrow$) | F1 score ($\uparrow$) |
| R-GCN | Distance + Distogram | 0.782 | 0.236 | **0.535** | **0.328** |
| | Distance + Distogram + correlation | **0.803** | **0.246** | 0.474 | 0.324 |
| R-GAT | Distance + Distogram | 0.814 | 0.289 | **0.598** | 0.390 |
| | Distance + Distogram + correlation | **0.819** | **0.293** | 0.584 | **0.391** |
| R-EGNN | Distance + Distogram | **0.861** | **0.376** | 0.606 | **0.464** |
| | Distance + Distogram + correlation | 0.843 | 0.339 | **0.610** | 0.436 |
| Binding Affinity Prediction (Mean number of nodes = 47) | | | | | |
| Model | Graph Type | MAE ($\downarrow$) | RMSE ($\downarrow$) | Pearson R ($\uparrow$) | Spearman R ($\uparrow$) |
| R-GCN | Distance + Distogram | **1.195** | **1.487** | **0.731** | **0.694** |
| | Distance + Distogram + correlation | 1.303 | 1.601 | 0.676 | 0.655 |
| R-GAT | Distance + Distogram | **1.176** | **1.487** | **0.740** | **0.713** |
| | Distance + Distogram + correlation | 1.319 | 1.630 | 0.661 | 0.630 |
| R-EGNN | Distance + Distogram | **1.275** | **1.560** | **0.699** | **0.674** |
| | Distance + Distogram + correlation | 1.359 | 1.672 | 0.632 | 0.614 |

Table 7: Effect of adding correlation edges to distance and distogram edges

Table 7 indicates that incorporating correlation edges in addition to distance and distogram edges yields little to no improvement in either binding site or binding affinity prediction, suggesting that the correlation information is largely captured by the existing distance and distogram relationships. This effect could be explained by the overlap between distograms and correlation edges, since both capture related aspects of the system's dynamic behavior.

