# OpenReview forum: "Learning Dynamic Protein Representations at Scale with Distograms"
_ICLR.cc/2026/Workshop/LMRL — ICLR 2026 Workshop LMRL Poster_

### Official Review · Reviewer_jz3X · 2026-02-11

**Rating:** 7
**Confidence:** 5

**Review:**

The authors employ Boltz-2 distograms to enrich graph constructions and predict both ligand binding site and binding affinity. They also predict mutation effects and RNA properties in absence of MD simulations. Limitations and future work are clearly stated and reasonable.

However, implying that distrograms reflect dynamics is an overstatement as they are typically linked with model uncertainty (as you correctly mention) and MSA-derived coevolutionary signals. Authors could show this by assessing the correlation with measures coming from Molecular Dynamics Simulations such as RMSF. Further, taking inspiration from the distance+correlation and distance+random edges scenarios, I wonder if part of the observed improvements may come from adding long-range message passing. A sensible sanity check would be to consider dense (weighted) graphs as a comparison. Authors should clarify that distrograms are model-dependent (Boltz-2 in this case) instead of data-driven, meaning that they inherit failure modes and biases coming from Boltz-2.

---

### Official Review · Reviewer_PZyk · 2026-02-25
**An Dynamic Encoding Approach, Some Experimental Conclusions Need Strengthening**

**Rating:** 5
**Confidence:** 4

**Review:**

Protein function often depends on conformational dynamics, but most ML methods rely on static structures, and the main way to obtain dynamic information — molecular dynamics simulation — is computationally expensive. This paper uses distograms from structure predictors to encode protein dynamic information as a cheap alternative, and integrates them into relational GNNs for prediction tasks.

**Pros**:
- Clear motivation; using distograms as a cheap proxy for dynamic information is a valuable idea
- Extension to RNA is an interesting exploration

**Cons**:
- The machine learning contribution is limited, and the architecture design lacks novelty
- In the binding affinity task, using distogram edges alone (without edge features) performs slightly worse than random edges. The authors attribute this to the binding pocket graph already being dense, but this explanation is not convincing enough
- Distograms are pairwise marginal distributions and cannot explicitly capture higher-order multi-residue dynamic correlations.
- The method relies heavily on MSA quality; its robustness for orphan proteins or systems with poor MSA coverage remains unclear


**Question**:
For a given distogram with a broad distribution, is it caused by true conformational flexibility or model prediction uncertainty?

---

### Official Review · Reviewer_TZtG · 2026-02-25
**A solid workshop paper exploring new combinations of data and machine learning architectures**

**Rating:** 8
**Confidence:** 5

**Review:**

The authors propose to use distograms, computed from conformational ensembles generated by Boltz as a relatively computationally inexpensive proxy for protein dynamics as an input for relational GNNs. They evaluate this approach on downstream tasks that include binding site prediction, binding affinity prediction, mutation effect prediction, and RNA property prediction. They compare it against alternative approaches that use atomic motion correlation matrices from molecular dynamics simulations and purely distance-based methods.

$\textbf{Strengths}$

The core idea presented by the authors is very interesting and clearly computationally efficient. Using relatively cheap structure predictions to extract distograms is elegant and improves upon using distances from single conformations.

They show that their approach can perform better than those using molecular dynamics-derived data.

The author's empirical data is strong as they include a relatively broad range of tasks and also include random edges as a baseline. Furthermore, they include ablation experiments.

$\textbf{Weaknesses}$

The performance improvements of their approach are modest. However, I do not see this as an issue here, as this is otherwise a very interesting explorative study into using a new data modality (generated conformational ensembles).

While further weaknesses, e.g. missing an exploration of the influence of distogram quality on the performance in downstream tasks, are present, I do not believe these are relevant for a workshop paper, as the article, as is, already merits discussion.

Overall, I believe this work to be timely and well-structured. It explores new ways to exploit and combine existing data and existing architectures, which is valuable.

---

### Meta-Review · Area_Chair_ytws · 2026-02-28

**Recommendation:** Accept (Poster)
**Confidence:** 4

**Metareview:**

Accept

---

### Decision · Program_Chairs · 2026-03-02

**Decision:**

Accept (Poster)

**Comment:**

Please see the meta-review.